# Single-cell replication profiling to measure stochastic variation in mammalian replication timing

Vishnu Dileep[1] & David M. Gilbert[1]

Mammalian DNA replication is regulated via multi-replicon segments that replicate in a defined temporal order during S-phase. Further, early/late replication of RDs corresponds to active/inactive chromatin interaction compartments. Although replication origins are selected stochastically, variation in replication timing is poorly understood. Here we devise a strategy to measure variation in replication timing using DNA copy number in single mouse embryonic stem cells. We find that borders between replicated and unreplicated DNA are highly conserved between cells, demarcating active and inactive compartments of the nucleus. Fifty percent of replication events deviated from their average replication time by ±15% of S phase. This degree of variation is similar between cells, between homologs within cells and between all domains genomewide, regardless of their replication timing. These results demonstrate that stochastic variation in replication timing is independent of elements that dictate timing or extrinsic environmental variation.

[1] Department of Biological Science, Florida State University, 319 Stadium Drive, Tallahassee, FL 32306, USA. Correspondence and requests for materials should be addressed to D.M.G. (email: gilbert@bio.fsu.edu)

In mammalian cells, large chromosome domains (replication domains; RDs) replicate at different times during S-phase, linked to chromatin architecture and genome integrity[1,2]. Although single DNA molecule studies have demonstrated that replication origins are selected stochastically, such that each cell is using a different cohort of origins to replicate their genome[3–8], replication timing is regulated independently of origin selection[9], and evidence suggests that replication timing is conserved in consecutive cell cycles[10–12]. However, measurements of replication timing in consecutive cell cycles have been limited to cytogenetic studies[10–12] and molecular methods to measure replication timing have been limited to ensemble averages in cell populations[13]. More recently, it has been shown that RDs correspond to structural units of chromosomes called topologically associating domains (TADs)[14]. TADs in close proximity replicate at similar times, segregating into separate higher order spatial compartments consisting of early replicating/active vs. late replicating/inactive chromatin[2]. Hence, quantifying the extent of cell-to-cell variation in replication timing is also central to understanding the relationship between large-scale chromosome structure and function. Here we use DNA copy number variation (CNV) to measure replication timing in single cells at different stages in S phase. By measuring the variation in replication timing, we find similar stochastic variation between cells, between homologs within each cell, and also between all domains genomewide, regardless of their time of replication in S phase. The borders separating replicated and unreplicated DNA are conserved between single cells and demarcate the active and inactive compartments of the nucleus. Overall, these results demonstrate that stochastic variation in replication timing is independent of extrinsic environmental factors as well as the mechanisms controlling the temporal order of replication.

## Results

**Single-cell replication measured using CNV.** Single-cell DNA copy number can distinguish replicated DNA from unreplicated DNA[15,16]. Specifically, regions that have completed replication will have twice the copy number compared with regions that have not replicated. Hence, we reasoned that measurements of DNA copy number in cells isolated at different times during S-phase could reveal replication-timing programs in single cells. Moreover, to separately evaluate the extent of extrinsic (cell-to-cell) vs. intrinsic (homolog-to-homolog) variability in replication timing, we examined both the differences in replication timing between haploid H129-2 mouse embryonic stem cells (mESCs) and the differences between maternal and paternal alleles in diploid hybrid *musculus* 129 × *Castaneus* mESCs that harbor a high single-nucleotide polymorphism (SNP) density between homologs, permitting allele-specific analysis. To generate single-cell CNV profiles, we used flow cytometry of DNA-stained cells to sort single S-phase cells into 96-well plates followed by whole genome amplification (WGA). Amplified DNA from each cell was uniquely barcoded and sequenced (Fig. 1a)[17,18]. Read counts of all cells were converted to reads per million (RPM) to control for variable sequencing depth. To control for amplification and mappability biases, we also sorted G1 and G2 cells, which contain a relatively uniform DNA content. Regions of low mappability and over amplification were removed based on the G1 and G2 controls. Read counts were normalized by dividing the coverage data of each single cell by the coverage of the G1 and G2 control cells. Next, a median filter was applied to smooth the data, producing CNV profiles in 50 kb bins (Methods).

We generated single-cell sequencing data for 199 mESCs, composed of 92 haploid H129-2 and 107 129 × *Castaneus* diploid mESCs. As we expected the CNV profile of mid-S-phase cells to

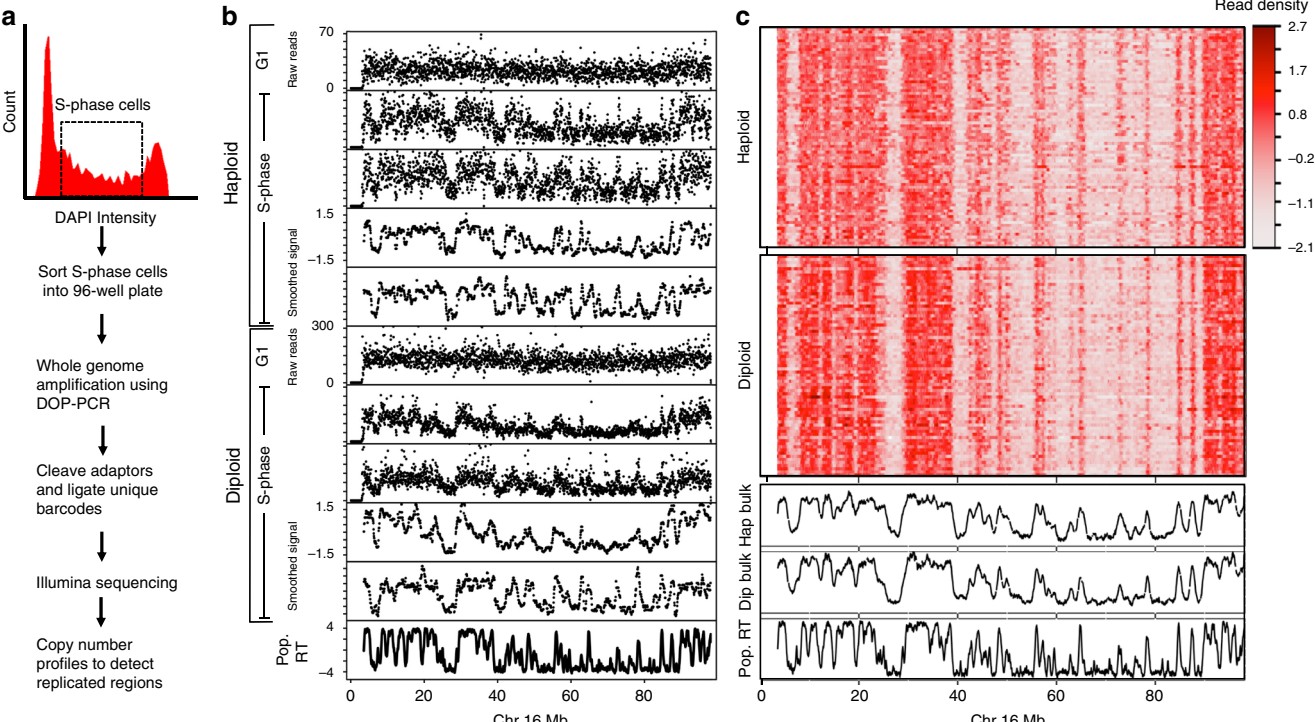

**Fig. 1** Single-cell replication using copy number variation. **a** Method for generating single-cell CNV profiles. **b** Representative single-cell CNV profiles of G1 and S-phase cells in both haploid and diploid hybrid cells. CNV profiles are shown as raw read count in 50 kb bins and after smoothing and corrections. **c** Heatmap of all single-cell CNV profiles after smoothing and corrections. The bottom three panels show aggregate of haploid single cells, aggregate of diploid single cells, and replication timing measured using population-based Repli-seq in the diploid hybrid cells

distinguish the maximum number of early and late replicating domains, we sorted a majority of the cells from mid-S-phase. We extended the sorting gates for the haploid H129-2 cell line to early and late S-phase (Supplementary Fig. 1a). Cells with few reads (5.5% of cells) or cells with complex karyotype aberrations/ complete loss of a chromosome (7% of cells) were discarded, whereas cells with aneuploidy were corrected by normalizing to the mean read density for those chromosomes in the control G1 and G2 cells (see Methods). Approximately 27% of the diploid hybrid cells showed karyotype aberrations or aneuploidy, with the most frequent being X-chromosome loss, consistent with previous observations[19,20]. Only 6% of haploid cells showed karyotype aberrations or aneuploidy (Supplementary Fig. 2).

Cells sorted from the middle of S-phase showed an oscillating CNV relative to the flat CNV profile for cells sorted from G1 or G2 phase (Fig. 1b). Regions with higher copy number aligned with early replicating domains detected by population-based replication timing data measured using Repli-seq[21]. Correcting the raw read counts for mappability and WGA biases, followed by smoothing and scaling the data, produced CNV profiles with reduced noise without any loss of information (Fig. 1b). Population-based replication timing profiles have revealed chromosomal segments that replicate relatively synchronously, appearing as a plateau in RT profiles. These segments are termed Constant Timing Regions (CTRs)[2]. Heat maps of single-cell replication-timing profiles from all single cells show clear signatures of domains that correspond to CTRs, indicating conservation of replication timing at the single-cell level (Fig. 1c and Supplementary Fig. 3). Average replication-timing profiles generated from both the haploid and the diploid cells show a Pearson's correlation of ~ 0.89 with population-based Repli-seq profiles, demonstrating the robustness of our protocol and computational pipeline.

Haploid and diploid cells were processed differently. For haploid single cells, replication data can assume two states, replicated or unreplicated. Hence, we binarized the corrected data to reflect the replicated vs. unreplicated states as follows. First, we segmented the smoothed CNV profiles to identify segments with higher and lower copy number. Next, for each cell, we generated several binary signals using evenly spaced threshold values. Finally, we chose the threshold value at which the distance (Manhattan) between the binary signal and the unbinarized segmented data was minimum (Methods and Supplementary Movie 1). Binary classification allowed us to rank individual cells within the spectrum of S-phase progression based on the number of bins identified as replicated. The ranking accurately reflected the early, middle or late fluorescence-activated cell sorting (FACS) sorting gates used to collect the single cells (Supplementary Fig. 1b). Outlier cells that did not correlate with any other cells were removed from further analysis (Supplementary Fig. 4 and Methods). At the end of these data processing steps, 75 out of 92 haploid S-phase cells had passed all quality control measures and were used for further analysis.

In contrast to haploid cells, single-cell replication data from diploid cells is an average of two homologs. Therefore, diploid cells can exhibit a third state characterized by asynchrony between homologs where one homolog has replicated and the other remained unreplicated. As it was not possible to consistently and confidently identify this third state based on copy number alone, we took advantage of the high SNP density of our hybrid musculus 129 × Castaneus mESCs to parse the diploid cell genomes into maternal and paternal haploid genomes and generate homolog-specific single-cell data by identifying regions of homolog asynchrony. We first segmented the smoothed CNV profiles and binarized them as described for the haploid cell data. Next, we generated homolog-specific coverage data by parsing the

sequencing reads based on homolog-specific SNPs. This allowed us to calculate an asynchrony score to measure homolog-to-homolog variability for each segment as the log ratio of 129 to Castaneus read coverage. As regions of low SNP density have fewer sequencing reads after parsing and we had variable total reads per cell, we filtered for those segments with high confidence of replication asynchrony as follows. First, we only considered segments with a minimum read number of 40 based on previous empirical determination of read number requirement for efficient copy number identification[18]. Next, the threshold for the asynchrony score was set to be outside the distribution of asynchrony scores calculated from the control G1 and G2 cells (Supplementary Fig. 5). Homolog-specific binarized data were then generated by modifying the original binarized data to reflect segments with asynchrony within each cell (Methods). At the end of allele-specific parsing, 71 out of 107 diploid S-phase cells had passed all quality control measures and were used for further analysis.

The combined haploid and homolog-parsed diploid binarized data ranked by the position of each cell in S-phase (total DNA content) revealed domains with incomplete replication during early S-phase that progressively completed replication in cells ranked later in S-phase (Fig. 2a and Supplementary Fig. 6). These domains correspond to early replicating CTRs identified in population-based replication-timing profiles[2]. Comparison of the binary single-cell replication timing to simulated deterministic (where each cell follows the population-based replication timing accurately) vs. simulated random replication timing indicates a high degree of cell-to-cell replication-timing conservation (Fig. 2a and Methods). In addition, the binary replication signal in mid-S cells corresponded to early and late replication in population-based replication timing data (Fig. 2b). Further, as our data reveals that replication timing is well conserved at the single-cell level, we reasoned that in order to detect cell-to-cell variability in replication timing for any particular domain, we would need to capture cells at a time during S-phase very close to the average replication time of that domain. In other words, replication-timing variability in early, middle, and late S-phase should be maximum for early, mid, and late replicating regions, respectively. To test this prediction, we calculated the variability of binarized data for all pairwise combinations of single cells that were ranked within one percentile of S-phase progression. Consistent with our prediction, we found a very pronounced increase in replication timing variability at segments whose population-based replication timing corresponded to the position of the single cell in S-phase (Fig. 2c).

**Variation in single-cell replication timing**. To quantify variability in replication timing during S-phase, we first converted the population-based replication timing in $\log_2$ enrichment scale to time in hours assuming a 10 hr S-phase[22] so that each 50 kb bin would have an expected average time of replication in hours. We then converted the percentage progression of each single cell in S-phase into hours during S-phase. Using these two quantities we generated a cell-specific "time from scheduled replication" profile by subtracting the number of hours the individual cell has progressed in S-phase from the population-based genome-wide replication time in hours for each 50 kb bin (Supplementary Fig. 7). Thus, for each cell, positive values indicate bins that are scheduled to replicate in the future and negative values indicate bins that should have already replicated.

To estimate the cell-to-cell extrinsic variability in replication timing, we calculated the fraction of cells that replicated each bin position within each possible "time from scheduled replication" in intervals of 0.1 h. For example, a mid-replicating position can only contribute bins with "time from schedule replication" within ± 5 h,

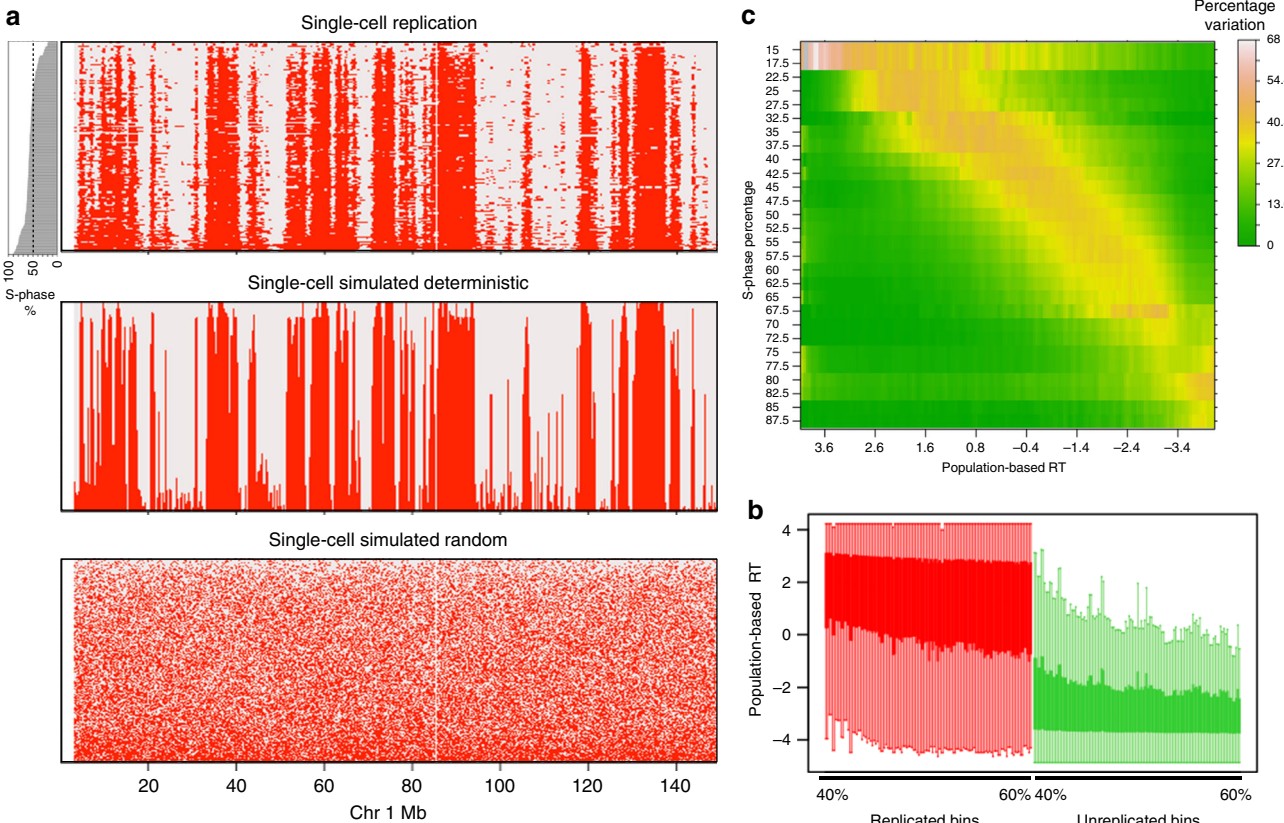

**Fig. 2** Binarized single-cell replication. **a** Binarized replication status in all haploid single cells and homolog-parsed diploid cells. The cells are ranked by their progress in S-phase, which is plotted as a bar plot on the left. The bottom panels show simulated deterministic and random replication for the identical S-phase distribution of single cells (Methods). **b** Boxplot of population-based replication timing for replicated (red) and unreplicated (green) bins for each single-cell ranked between 40 and 60% S-phase progression. **c** Heatmap of variability between pairs of cells ranked within one percentile of each other. The Y axis is average S-phase rank of pairs of cells (all pairs ranked within 1% of S phase), measured in intervals of 5% of S-phase progression with a step size of 2.5. The x axis is the replication timing (RT) measured by population-based repli-seq, measured in intervals of 0.1 with a step size of 0.05. The pairwise variability between cells (measured using the binarized data) is the percentage of 50 kb bins where there is a transition in the binary signal for the given S-phase progression interval and population-based RT interval

whereas an early replicating region that replicates at 2 h into S-phase can only contribute bins with a "time from schedule replication" from −8 to +2 h (total S-phase length is 10 h). As this calculation was done for all 50 kb bin positions in the genome, we plotted the mean across all bins for each 0.1 h interval of "time from scheduled replication". Supplementary Figure 8 explains this calculation at an exemplary location in the genome. The kinetics resembled a sigmoid curve that was consistent with previously described theoretical models of stochastic replication-timing regulation based on population-based replication timing data[7,23–25] (Fig. 3a and Supplementary Fig. 9). To estimate the stochasticity, we modeled the kinetics using standard parameters of a sigmoid curve (Methods). Then we estimated $T_{width}$, defined as time it takes to progress from 25% to 75% of cells replicated. $T_{width}$ was 2.7 h, which is much lower than the 10 h S-phase, consistent with a stochastic model of DNA replication[7,24].

To calculate the intrinsic (within cell) replication kinetics, we performed a similar analysis on each haploid cell and homolog-parsed diploid cell independently. We calculated the fraction of bins that were replicated for each possible "time from scheduled replication" in intervals of 0.1 h across a single genome. The within cell kinetics for all cells was very similar to the cell-to-cell kinetics with an aggregate $T_{width}$ of 2.5 h (Fig. 3b). In contrast, the randomized control had a much larger $T_{width}$ of 15.5 h and 13.7 h for intrinsic and extrinsic, respectively (Fig. 3a, b). Finally, we compared the homolog-to-homolog variation in the diploid cells

to "time from scheduled replication". Population Repli-seq has revealed high conservation of replication timing profiles between the homologs with only 12% of the genome showing detectable genome-specific variation[26]. Homolog-to-homolog variation was measured as percentage of bins that differ between homologs (absolute difference between the binary signal for all homologous pairs of bins). We limited the analysis to 10 cells with the highest read number after homolog specific read parsing. As expected, the maximum variation is at the regions that are actively replicating at the time of collection (time from scheduled replication = 0 h) (Fig. 3c). The cumulative sum of the variation resembles a sigmoid curve, very similar to the cell-to-cell variation ($T_{width}$ = 3.1 h). These results demonstrate that extrinsic and intrinsic variation in replication timing are indistinguishable, favoring a model of replication timing regulation where the timing is the outcome of stochastic origin firing and is not affected by the precise environment within a cell. Intriguingly, we could not detect any difference in stochasticity between early and late replicating bins (Fig. 3d), demonstrating that the mechanisms determining the scheduled replication time of any given domain are independent from those determining probability of a domain firing at its scheduled time.

**Conservation of active/inactive compartments in single cells.**
Population-based replication timing data reveals that CTRs are

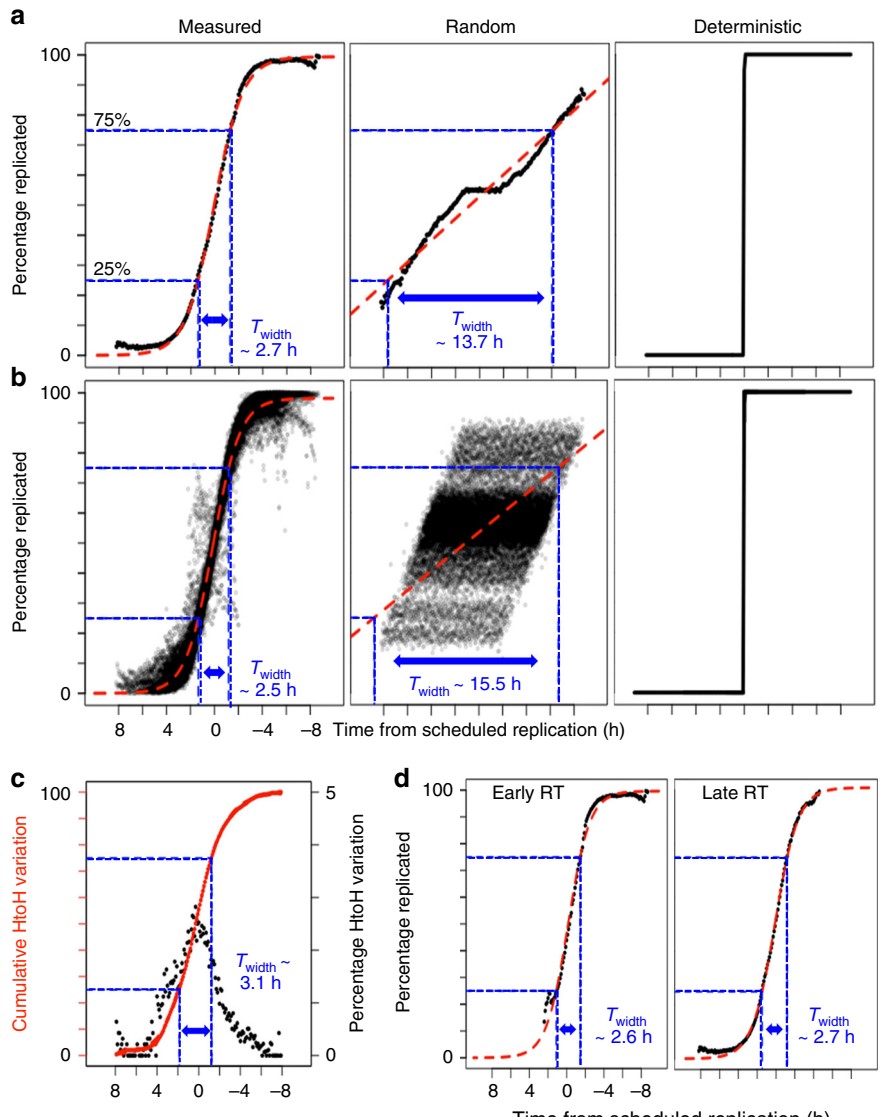

**Fig. 3** Measuring single-cell replication variability. **a** Cell-to-cell variability vs. "time from scheduled replication" in hours. The mean across all 50 kb bin positions is plotted for each 0.1 h interval on the x axis and the red line is the sigmoid fit. Control models of random vs. deterministic replication are shown for comparison. **b** Within cell variability across the genome plotted for each single-cell independently, similar to **a**. **c** Homolog-to-homolog variability vs. "time from scheduled replication". The black scatter plot measures percentage of bins that show homolog asynchrony (right y axis) for each 0.1 h interval on x axis. The overlapping solid and dotted red line are the cumulative sum of the variability and it's sigmoid fit, respectively (left y axis). **d** Cell-to-cell variability for early (RT > 0) vs. late replicating regions (RT < 0)

punctuated by segments of DNA that replicate gradually later with time, termed timing transition regions (TTRs)[2]. The binarized single-cell data from mid-S cells reveals conserved locations of copy number shifts that align with TTRs observed in the population-based data (Fig. 4a). The aggregate alignment of TTR centers (mid-transition) to the closest single-cell copy number shifts reveals a strong enrichment of the copy number shifts near the population-based TTR centers (Fig. 4b). In principle, the positions of copy number shifts should show maximum distinction between early and late replicating CTRs in cells that were in mid-S phase. Consistent with this, the alignment of single-cell copy number shifts to TTRs is highest in cells ranked closer to the middle of S-phase (Fig. 4b). The randomized control does not show any enrichment whereas the deterministic simulation shows near perfect enrichment at TTRs in mid-S phase cells. Thus, the regions of timing transition from early replication to late replication are conserved in single cells.

Chromatin conformation capture studies using Hi-C have revealed the broad segregation of chromatin into two functionally distinct compartments (A and B) that correspond to active and inactive chromatin[27]. This A/B compartmentalization strongly correlates with spatially and temporally distinct early/late replicating CTRs, respectively[28,29]. To measure the prevalence of these compartments in single cells, we used mid-replicating cells to construct a matrix representing co-regulated bins (pairs of bins that are both replicated or both unreplicated) as red and oppositely regulated (one replicated one not) regions as blue. The co-regulated regions were conserved between cells and recapitulated the nuclear compartmentalization measured by Hi-C with remarkable accuracy (Fig. 4c). Next, we calculated a matrix of pairwise absolute distance for all possible pairs of 50 kb bins within each chromosome, using the binary replication status across all cells. This heat map matrix shows regions of coordinated replication in single cells and was almost identical

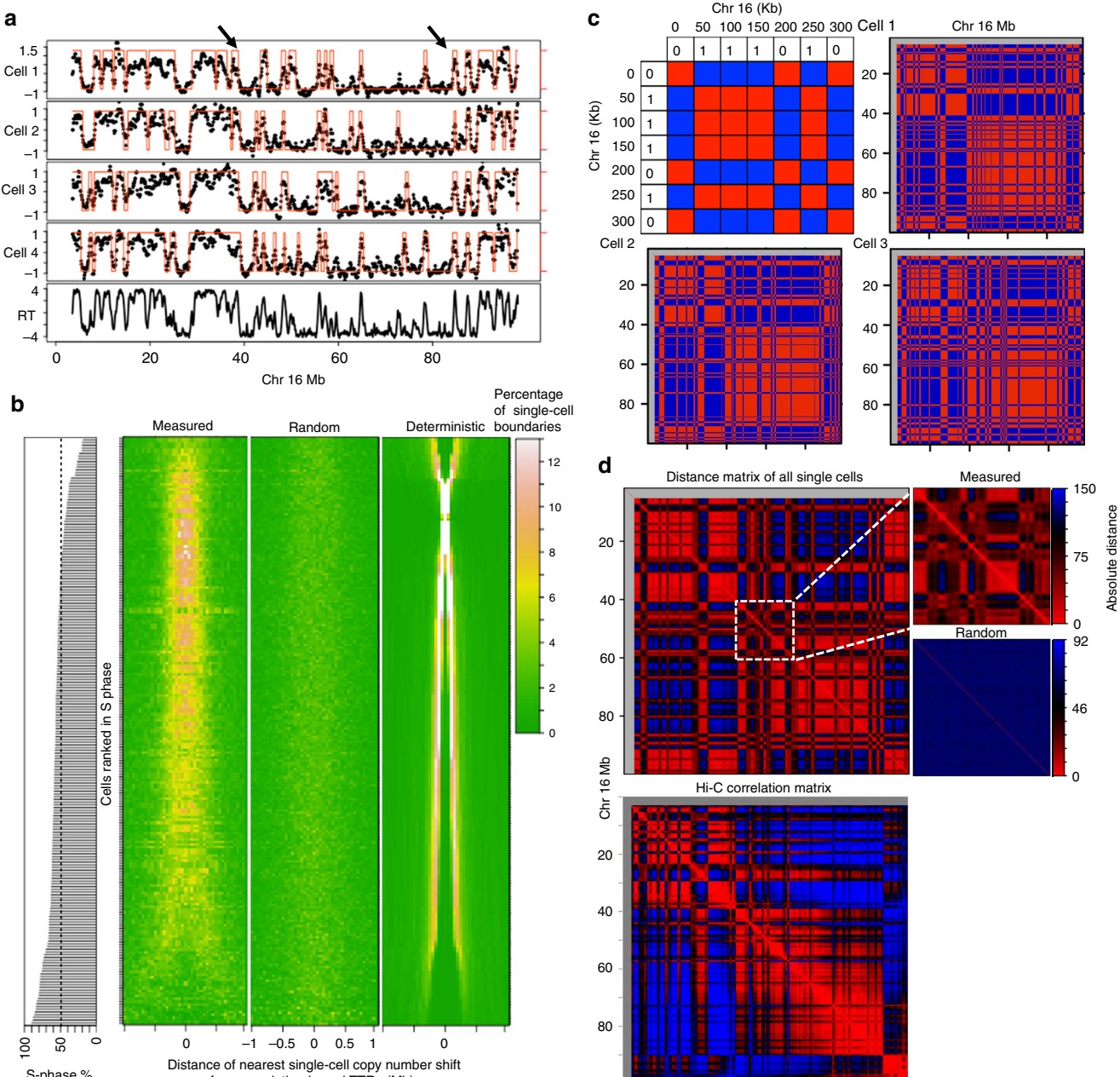

**Fig. 4** Conservation of TTRs and nuclear compartments in single cells. **a** Binarized single-cell data overlaid on smoothed single-cell CNV profiles. The arrows indicate representative TTRs in population-based data that align with copy number shifts across single cells. **b** Heatmap showing the distance of the closest single-cell copy number shift from all population-based TTR centers. Each row represents one single-cell and the heat map shows the percentage of closest single-cell copy number shifts in that cell at different distances from the TTR center. The cells are ranked by position is S-phase. **c** Coordinately regulated bins in each single-cell are color coded red and oppositely regulated bins are color coded blue as shown in the top-left schematic. Whole chromosome plot of single cell from mid-S-phase shows coordinated replication across cells and align with Hi-C compartments (shown in **d**). **d** Plotting pairwise absolute distance for all possible pairs of 50 kb bins within each chromosome, using the binary replication status across all cells reveal the functional compartmentalization of replication that show strong similarity to Hi-C compartments plotted using Juicebox[32]. The randomized control of single-cell replication lacks compartments highlighted by plotting a 20 Mb segment of Chr 16

to the Hi-C correlation matrix. These results show the conservation of functional (replication timing) nuclear compartments from cell-to-cell that correspond to the structural compartments measured by Hi-C (Fig. 4d).

In summary, we used CNV to measure replication timing in single cells in both haploid and diploid cells. The results support a model in which intrinsic variability and extrinsic cell-to-cell variability are similar, regardless of the timing or chromatin state of each domain, with most individual domains replicating within

± 15% the length of S-phase from their scheduled time (Fig. 3a). As our data suggest that replication timing is considerably less stochastic than replication origin selection, it will be important to investigate whether the firing potential of origins within each RD is integrated to produce a more deterministic replication-timing program. Future studies will be necessary to reveal the molecular events contributing the degree of stochasticity. Here we have developed methods to measure this degree of stochasticity. We also show that the locations of early to late timing transition

observed in population-based replication timing data are conserved in single cells. It has been previously shown that early and late replicating segments observed in population-based replication timing data align with the Hi-C based A and B sub-nuclear compartments, respectively[28,29]. A recent study revealed the conservation of the A and B compartments at the single-cell level[30]. The regions of the genome that are replicated at similar times in single cells align with these compartments demonstrating their structural and functional conservation. Overall, our results show that the spatio-temporal DNA replication program is conserved in single cells, and provides a direct quantification of single-cell replication timing variability, supporting a stochastic model of replication-timing regulation.

## Methods

**Cell culture and sorting.** Diploid F121-9 hybrid cell line and Haploid H129-2 (ECACC 14040205, a gift from Martin Leeb) mESCs were cultured in feeder-free 2i media. To sort single nuclei, one million cells were suspended in 0.5 ml of NST-DAPI buffer. NST buffer was made by mixing the following components in ddH$_2$O for a final volume of 800 ml: 146 nM NaCl, 10 mM Tris base (pH 7.8), 1 mM CaCl$_2$, 21 mM MgCl$_2$, 0.05% (wt/vol) bovine serum albumin and 0.2% (vol/vol) NP-40. Then, 200 ml of 106 mM MgCl$_2$ and 10 mg of DAPI was added to 800 ml of NST buffer to make NST-DAPI buffer. Next, we used the FACS AriaII flow cytometer to sort single nuclei from G1, S, or G2 phase into a 96-well plate contain lysis buffer (Sigma SeqPlex, SEQXE-50RXN).

**Population replication-timing profiling.** Genome-wide population replication timing was measured using Repli-seq protocol[21]. Briefly, synchronously cycling cells were pulse labeled with the nucleotide analog 5-bromo-2-deoxyuridine (BrdU) for 2 h to mark nascent DNA. The cells were sorted into early- and late S-phase fractions (20,000 cells per fraction) on the basis of DNA content using flow cytometry. Genomic DNA was isolated from each fraction using Zymo Quick-DNA Microprep kit (D3020). Next, the genomic DNA was fragmented using the Covaris E220 sonicator to get 200 bp average fragment size. The sheared DNA was end-repaired and illumina sequencing adaptors were added using NEBNext Ultra DNA Library Prep Kit for Illumina (E7370). Next, BrdU-labeled DNA from each fraction was immunoprecipitated using 0.5 μg of mouse anti-BrdU antibody (BD 555627) followed by 20 μg rabbit anti-mouse IgG (Sigma M7023). The immuno-percipitated pellet was digested overnight with Proteinase K and purified using Zymo DNA Clean & Concentrator −5 (D4003). The purified nascent early and late S-phase fraction DNA was differentially indexed and amplified using the NEBNext Multiplex Oligos for Illumina (E7600S). Amplified early and late fractions were sequenced on a Hiseq2500 platform using the 50-cycle single end format. Reads from each fraction was converted to RPM. Population-replication timing was then measured as the log$_2$ enrichment of early reads over late reads for each 50 kb bin position across the genome.

**Single-cell sequencing.** Single-cell sequencing was performed as described in two previous reports[17,18]. Briefly, the cells were lysed and amplified using the Sigma SeqPlex kit. The WGA products were purified and the WGA universal adaptor sequences were removed as per the kit protocol, leaving NN overhangs. Next, unique barcoded Illumina adaptors with NN overhangs (gift from Timour Baslan and James Hicks) were ligated to the products from the previous step. The previously published barcoded oligos are provided in the Supplementary Data 1[18]. Upto 96 uniquely barcoded cells were pooled together and amplified. Samples were then quantified using the Bioanalyzer and quantitative PCR, and sequenced on Hiseq 2500 sequencer using 50 bp single-end read format. Eleven cells were sequenced without restriction enzyme-based WGA universal sequence removal[17]. These samples were sequenced at 100 bp read length to get sufficient mappable reads after adaptor sequence trimming.

**Read mapping.** Reads were demultiplexed based on their unique barcodes. Both cast/129 and H129-2 reads were mapped to mm10 mouse genome assembly using Bowtie 2 with default parameter settings. Reads with Mapping Quality score of above 10 were retained for further analysis. PCR duplicates were removed using rmdup tool in samtools. Mapped reads were binned into 50 kb windows. Homolog-specific read mapping based on SNPs was done using a previously published pipeline[21].

**Data correction and smoothing.** Single-cell sequencing data binned into 50 kb windows were used for the data correction and smoothing. Single cells with less than 250,000 reads were discarded and reads were then converted to RPM. Cells with complex karyotype aberrations or complete chromosome loss were discarded while cells with aneuploidy were corrected by normalizing to the mean read density of those chromosomes in the control G1 and G2 cells. Complex karyotype

abnormalities which were found to occur in very few cells were identified by plotting the whole-genome coverage at 1 Mb bins.

G1 and G2 cells showed a relatively uniform coverage profile as expected and thus were used to account for GC bias, WGA bias, and to discard regions with mappability issues. The mean of 5 G1 and cells and 1 G2 cell was calculated for all 50 kb bins. Bins with extreme values (mean RPM > 99 percentile and mean RPM < 1 percentile) were identified and masked in all single cells. To identify repetitive segments of the genome with low mappability, we segmented the mean G1/G2 coverage using Piecewise Constant Fits (PCF) in R using package "copy number"[31]. The parameter used were γ = 3 and kmin = 10. Segments with mean RPM lower than 5 percentile were discarded. Finally, all single cells were divided by the mean G1/G2 coverage.

Next, each single-cell data were centered and scaled to have an equal interquartile range. Extreme values (mean RPM > 99 percentile and mean RPM < 1 percentile) in each single-cell data were removed followed by median smoothing with a span of 15 windows.

**Data segmentation and binarization.** The corrected smoothed data was segmented using PCF (R copynumber package) to identify segments with similar copy number. The parameters used were γ = 3 and kmin = 5. Next, the segmented data at 50 kb resolution was used to binarize the domains as replicated or unreplicated. Accurate binarization of the segmented data depends on choosing the correct threshold for each single cell separately. To this end, we used a brute-force strategy and applied 100 equally spaced thresholds spanning the distribution of the segmented data. At each step the segmented data was binarized and finally the best binary fit was chosen.

The binary fit was calculated using manhattan distance between the binarized data and the unbinarized segmented data. Historically, 1 and 0 are used to denote binary data. However, to calculate the similarity based on Manhattan distance, the binary values and the segmented data must have similar magnitude. To do this, we first used mixture model fitting with two components (normalmixEM function in R package mixtools) to identify the replicated and unreplicated fractions in the segmented data. Then the binary values were set as the mean of the components. On the rare occasion when component means were very close (< 0.7) the binary values were decided based on the skew of the segmented data. This happened predominantly when the cells were in very early or very late S-phase, as the fraction of replicated segments and un-replicated segments respectively were too low for the mixture model to identify as a distinct component. A positive skew (skew > 0.2) indicated cells that are in early S-phase and the binary values were set to 50 percentile and 95 percentile of the segmented data. A negative skew (skew < − 0.2) indicated cells that are in late S-phase and the binary values were set to 5 percentile and 50 percentile of the segmented data. Otherwise, the binary values were set to 25 percentile and 75 percentile of the segmented data. The binary signal with the minimum manhattan distance (highest similarity) from the segmented data was chosen as the best fit (Supplementary movie 1). Outlier bins with a segmented value outside of ± 2 were not used for the threshold calculation. Sixty-seven cells had at least one outlier bin and the average amount of outlier bins was 0.1% of the genome.

**Homolog-specific binary signal.** Homolog-specific binary signal for diploid hybrid cells were generated based on the homolog parsed sequencing data. First, we used un-parsed data to first segment and then binarize as described above. Two, identical copies of the binary signal were generated and assigned one to 129 allele and the other to the Castaneus allele. The binary signals from each allele can then be modified for segments that are identified to be asynchronous. To identify asynchronous segments, we calculated an asynchrony score (R) for each segment as:

$$R = \log 2(\text{reads in 129 allele}/\text{reads in Castaneus allele}) \qquad (1)$$

Theoretically, a domain that is completely replicated in one homolog and completely unreplicated in the other homolog will yield an R score of 1 or − 1. However, this ideal value cannot be expected due to two reasons: (1) owing to the sparse nature of single-cell data, the large domains identified as replicated or unreplicated by the segmentation and binarization may still contain small segments that differ in their replication status; (2) the distribution of SNPs is non-homogenous resulting in loss of coverage after parsing.

Therefore, we discarded segments with a total read number is < 40. This number was determined based on a previous study that empirically determined total number of reads required for efficient copy number analysis[18]. Next, the threshold for the asynchrony score was set to be outside the distribution of asynchrony scores calculated from the control G1 and G2 cells. The thresholds used were R > 1.11 for 129 allele to be classified as replicated and R < − 1.18 for the Castaneus allele to be classified as replicated (Supplementary Fig. 5).

**Removing outlier cells.** Outlier cells were defined as cells that don't correlate with any other cells after binarization. A heat map of genome-wide manhattan distance using binarized data for every pairwise combination of haploid cell ordered by their rank in S-phase reveals cells that are ranked close together have the least distance and cells ranked far apart in S-phase have maximum distance. But some cells have low similarity to all the other cells and appear as streaks in the heat map

(Supplementary Fig. 3). These cells were removed from further analysis. In total, 19 outlier cells were removed from diploid data and 5 outlier cells were removed haploid data.

**Calculation of deterministic and randomized model.** The deterministic model assumes that each cell follows the population-based replication-timing profile precisely. For a given single cell with 'x' percentage of the genome replicated, the deterministic profile is given by assigning all 50 kb bins with average population-based timing greater 'x' percentile as replicated. To generate the random profile, 'x' percentage of 50 kb bins selected randomly are assigned as replicated.

**Calculating sigmoid fit.** The sigmoid fit was modeled using a non-linear least squares approach (function *nls* in R) using the formula:

$$y(x) = \frac{K}{1 + e^{-B(x-M)}} \tag{2}$$

where, $K$ is the maximum value of the sigmoid, $B$ is the growth rate, and $M$ is the $x$ intercept at the sigmoid's mid-point. The initial values of $K$, $B$, and $M$ were set to 100, 0.7, and 0, respectively.

**Code availability.** All R code used in this manuscript is available upon request.

**Data availability.** All data sets generated in this study are deposited in the NCBI Gene Expression Omnibus (GEO; http://www.ncbi.nlm.nih.gov/geo/) under the accession number GSE102077.

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

## Acknowledgements

We thank T. Baslan and J. Hicks for providing indexing barcodes; M. Leeb for providing the haploid cell line; J. Bechhoefer, B. van Steensel, and A. Belmont for their helpful comments. This work was supported by NIH GM083337, GM085354, and DK107965 to D.M.G.

## Author contributions

V.D. and D.M.G. conceived the study. V.D. performed the experiments and the analysis. V.D. and D.M.G. wrote the manuscript.

## Additional information

**Competing interests:** The authors declare no competing financial interests.

