## [Peer Review File · Nature Communications]

Reviewers' comments:

Reviewer #1 (Remarks to the Author):

Comments for transmission to the authors

The manuscript from Dileep and Gilbert described a new method permitting to characterize the variation in replication timing program from cell-to-cell or between homologous chromosomes. The main conclusion is that there is no difference (or a little bit) in replication timing program between cells and homologous

Generally speaking, this manuscript is sorely lacking in explanation and clarification, and the wording is, from my point of view, imprecise, in some cases with a lack of rigor. However, the approach is technically very interesting and I think it may be important, or even essential, for certain biological issues addressed in the future.

1. Title

Can you explain or give us your definition of "stochastic" What is your point of view when you used this term... mathematical thing? I think that your results reveals stochastic replication timing profiles from cell-to-cell, not a stochastic regulation

For me, this title is too strong compared to the results produced in this manuscript

2. Abstract

I would like you to explain why cell-to-cell variation is central, is it not too excessive?

3. Paragraph #1

Exactly the same sentences as in the abstract. This is not acceptable for this level of publication and again why variation is central? You can more develop in this introduction if it is one because there is no plan in this manuscript.

4. Paragraph #2

G1 and G2 does not contain uniform DNA. Why?

- some papers (and certain from your lab) indicate that you have replication during G1 and G2 fraction

- some parts of the genome could be duplicated or deleted

For me, the normalization explanation is not clear and Why you used a 50kb windows? You must give justifications and in suppl. Data, you can show different smooth with different length of bins

5. paragraph #3

You sorted a majority of the cells from mid-S-phase. How you know that the cells in this fraction have the same S phase length? Is it possible to show us the length of S-phase from several single cell to see if the length of S-phase is uniform from cell-to-cell. If it's not the case, you add bias in your analysis. We have an idea with the supp fig 1 and the figure b show a variation between 40% of percentage replicated and 70%. So the little variation that you observe later is probably due to this heterogeneity of the S-phase length from cell-to-cell.

Why you don't sequence again cells with few reads and add these new reads to the previous permitting to add these samples to the analysis?

So, finally, how many cells are discarded?

Paragraph #4

in fig1b, I would like to see a G2 profile and a replis-seq classical profile

6. paragraph #5

Is it possible to have a box plot of distribution of the signals for each chromosome, because if the signal range change from a chromosome to other, you introduce a bias during binarization.

Why you remove outlier bins with a segmented value outside of +/- 2? and how many you

removed?

You choose 100 equally spaced thresholds, why? Could you perform your Manhattan graph with 10, 50, 500 thresholds and indicate us which is the best?

I calculated that you removed 20% of haploid cells, is not it too much?

7. paragraph #6

You mentioned region with low density. How many? Distribution for each chromosome? Which type of regions, more in Early or in late or same proportion ?

The minimum number of reads is 40 chosen with an empirical determination. Are you sure that in your case it is adapted?

I calculated that you removed 35% of single diploid cells, is it not too much?

In general, I would like to see the same analysis with 10 single cells, 500 and 1 000. It is possible that with more single cells we can see more variation from cell-to-cell and between homologous chromosomes

8. paragraph #9

Again, can you determine the real S-phase length in your experiments?

9. paragraph#10

I would like to know if the homologue variations are more in Early or in late or in the same proportion.

10. last paragraph

you written "within +/- 15%" How do you measure that ?

In the sentence "overall, our results show..." there is two "THAT"

In method

For data correction and smoothing, you calculated the mean of 5 G1 and 1 G2. I would have preferred 5 G1 and 5 G2.

In Removing outlier cells paragraph, you indicate "some cells have low similarity..." How many exactly ?

Reviewer #2 (Remarks to the Author):

The authors present a detailed assessment of cell-to-cell variability in replication timing by utilizing high-quality single-cell DNA sequencing acquired from S-phase cells. Overall the manuscript is very well written and clear. The comparison to existing ensemble datasets suggest that the methods and analyses are robust, providing confidence in the quality of the study. The conclusions drawn from the analysis are well supported and justified.

In the second paragraph of the main text (pg4 top) where the authors briefly describe their workflow, they state that "Read counts were normalized by dividing the coverage data of each single-cell by the coverage of the G1 and G2 control cells." This statement implies no normalization for variable read counts for each cell or binning over windows was used, which is not the case when reading the methods (which are described well). Changing this sentence to suggest what was actually carried out would add clarity. (eg "Windowed read counts per million total reads

for each single cell were normalized by dividing..." etc...

Also "single cell" is a noun, "single-cell" is an adjective. In most instances it is used correctly, but some may have been missed or auto-corrected which happens frequently in my experience

Figure 1c: A label for what the heatmap scale is showing should be added instead of having to root through the legend. Similarly, there should be scale bars on the bottom portion.

I really appreciate the binarized analysis strategy. Very clear and seems to work well.

The authors attempt to investigate homologue-specific variability within the same individual cells, but appear to have been unable to due to the low haplotype-resolved read counts. (at least not in a meaningful way). While I appreciate this analysis, the power was not really there, so it may be worth removing the details from the main text and simply stating that it was attempted but coverage was too low and refer to the methods section. After continuing reading, the authors then come back to homologue-specific differences based on the domain-specific variability and see little difference. So they were able to say something meaningful with that analysis. Perhaps consolidating the homologue-specific analysis to this one section would help with the flow of the manuscript instead of earlier on where it appears nothing came of it then looping back to it.

Figure 2d-g could be lined up a bit better or even made a bit larger – these are plots that do a lot to convey important information.

Fig 3b include a title for the color scale bar.

P13 typo: "pairs of bind" should be "pairs of bins"

Correlation with HiC compartments is great.

Some of the conclusions are written in a contradictory way – ie "highly conserved, yet stochastic" but "far from random". I understand the conclusion that is being made, but it should be broken down into what you mean by the stochastic component (ie domains do not always activate at the same time/order in a replication origin selection context) and that when you refer to random you are implying no specified program, but genome-wide random which is not the case because actual timing is quite close.

Reviewer #3 (Remarks to the Author):

General

This submission provides a novel perspective of chromosome structure, probing into the flexibility of the replication timing program in the context of large, functionally distinct chromatin compartments. The paper reports the development and implementation of single-cell based analyzes of replication timing domains, demonstrating that the boundaries of chromatin regions that share the time of replication at a single cell level were similar to the boundaries of active and inactive chromatin compartments. Importantly, mapping the extent of genome duplication in single cells, and analyses of the extent of variations in single cell replication timing, allow an accurate interpretation of population-based replication timing data.

The results reported in the paper were obtained using novel methodologies to measure DNA copy number in individual cells. The studies were performed with sufficient resolution to distinguish between chromosomal zones of replicated and unreplicated DNA, providing a snapshot of the stage of replication completion (corresponding to cell cycle stage) in single cells. The paper provides compelling evidence that the replication timing program could be inferred from single-cell based

measurements. This is an important methodological development, likely to provide tools that will considerably advance the field of DNA replication.

The paper is written very clearly, presenting state-of-the-art experiments with appropriate controls. Single-cell copy number variation data are described with sufficient detail. The manuscript will be improved by the inclusion of a better description of the population-based (Repli-seq) data. This is important, because the relationships between the single-cell data and population-based data are critical for the variability analyses. Some other minor points require clarifications. Overall, however, this report represents an important contribution to the field.

Comments and minor suggestions:

1. The manuscript in its current form describes the methodology and primary observations for single-cell analyses in detail (for example, supplementary Figure 4), but the parallel report for Repli-seq does not provide the same level of detail. The primary reference for the replication timing data is an unpublished BioRxiv protocol. Although population-based replication timing analyses are less novel than the single-cell data, the comparison is important and both sets of data should be presented with sufficient detail. In addition, the current analysis relies on a novel variation of the Repli-seq technique, and the paper would benefit from a more detailed description of the results.
2. Related to the above, Page 7, last paragraph: "These domains correspond to early replicating CTRs identified in population-based replication-timing profiles". A citation, a reference to a figure, or a URL with the population-based data should be provided here.
3. To allow a full evaluation of the results, the paper should provide URLs for data access (single-cell and population based) in public depositories.
4. Analyzing the extent of variation based on the ratios of replication timing in the two mouse species is a clever and informative approach. However, it would be good to include an analysis of the general parameters underlying the variations reported in the paper (for example, it would be beneficial to report the size distribution of replication domains (CTRs and TTRs) in the two homologue-parsed chromosomal sets).
5. Related to the above, how does the variation in population-based replication timing domains in the current study relate to variations in chromatin compartments calculated from Hi-C in the hybrid and the parental species? Also, how does the extent of variation between homologues in the current study compare to the variation obtained in previous reports of replication timing variation (for example, in humans using a phased chromosome approach: Mukhopadhyay et al pubmed id 24787348)?
6. The manuscript in its current form reports data from hybrid cells on page 4, yet the benefits of using those hybrids to the study of differential replication timing variations are described in detail in page 6. It would be better to move the text from the last paragraph on page 6 to an earlier point in the manuscript before reporting the results.
7. Legend to Figure 1: "population-based BrdU-IP": if this is the same technique referred to as "Repli-Seq" in the text, it would be good to use a consistent term throughout the manuscript.

We thank the reviewers for a thorough reading of our manuscript. Below is a point by point response. Responses are in bold and changes in the manuscript have been underlined.

We were asked to provide the GEO links to our datasets, which we had provided in the original cover letter. We have now added a Data Availability statement in the manuscript and it is also pasted below for the reviewers' convenience:

All datasets generated in this study are deposited in the NCBI Gene Expression Omnibus (GEO; <http://www.ncbi.nlm.nih.gov/geo/>) under the accession number GSE102077 (secure token: qtopuymyvvidxsv) and will be made publicly available upon manuscript acceptance.

Reviewers' comments:

Reviewer #1 (Remarks to the Author):

Comments for transmission to the authors

The manuscript from Dileep and Gilbert described a new method permitting to characterize the variation in replication timing program from cell-to-cell or between homologous chromosomes. The main conclusion is that there is no difference (or a little bit) in replication timing program between cells and homologous

Generally speaking, this manuscript is sorely lacking in explanation and clarification, and the wording is, from my point of view, imprecise, in some cases with a lack of rigor. However, the approach is technically very interesting and I think it may be important, or even essential, for certain biological issues addressed in the future.

1. Title

Can you explain or give us your definition of "stochastic" What is your point of view when you used this term... mathematical thing? I think that your results reveals stochastic replication timing profiles from cell-to-cell, not a stochastic regulation

For me, this title is too strong compared to the results produced in this manuscript

The word ‘Stochastic’ is commonly used to imply a probabilistic model where certain regions have higher or lower probability of replicating at a given time as opposed to a deterministic program. The term “stochastic” has been widely used in previous theoretical models to define DNA replication, transcription, and other biological processes.

We have changed the title to remove the word “regulation”:

Single-cell replication profiling to measure stochastic variation in mammalian replication timing.

2. Abstract

I would like you to explain why cell-to-cell variation is central, is it not too excessive?

We have removed this statement and explained it more fully in the first paragraph of the text.

3. Paragraph #1

Exactly the same sentences as in the abstract. This is not acceptable for this level of publication and again why variation is central? You can more develop in this introduction if it is one because there is no plan in this manuscript.

We have now extensively modified this first paragraph to fully explain the gaps we are addressing in this manuscript.

4. Paragraph #2

G1 and G2 does not contain uniform DNA. Why?

We have changed the sentence to “relatively uniform”.

- some papers (and certain from your lab) indicate that you have replication during G1 and G2 fraction

By definition, G1 and G2 cells do not undergo DNA replication (with rare exceptions). The reviewer may be referring to the small numbers of contaminating S phase cells in the G1 and G2 sorted fractions, but this is a common limitation of synchronization.

- some parts of the genome could be duplicated or deleted

A fraction the cells do show duplications and deletions. We now show a supplementary figure (Supplementary Fig 2b) showing example cells that do exhibit duplications and deletions to avoid any confusion.

For me, the normalization explanation is not clear and Why you used a 50kb windows? You must give justifications and in suppl. Data, you can show different smooth with different length of bins

The bin size of 50kb was chosen for the following reasons:

- 1) It was a trade off between preventing bins with low read number vs retaining maximum resolution.**
- 2) Also we wanted the bin size to be sufficiently lower than average size of replication domains (400-800kb) that are the units of replication timing regulation.**

Now we show a supplementary figure (Supplementary Fig. 9) where haploid cells were binned at 25 kb, 50 kb, 100 kb, 150 kb and 200 kb, revealing bin size does not affect our conclusions.

5. paragraph #3

You sorted a majority of the cells from mid-S-phase. How you know that the cells in this fraction have the same S phase length? Is it possible to show us the length of S-phase from several single cell to see if the length of S-phase is uniform from cell-to-cell. If it's not the case, you add bias in your analysis. We have an idea with the supp fig 1 and the figure b show a variation between 40% of percentage replicated and 70%. So the little variation that you observe later is probably due to this heterogeneity of the S-phase length from cell-to-cell.

The length of S phase is internally normalized in our study by always directly measuring the total percentage of DNA replicated per cell and the variation seen in Supplementary Fig 1 is the variation in the position of the cell in S-phase based on percentage of genome replicated. The population replication timing is also all relative to the percentage of DNA replicated. Thus, there is no bias due to either time or FACS sorting. We have added more details to the methods section on population repli-seq that may clarify this confusion.

Why you don't sequence again cells with few reads and add these new reads to

the previous permitting to add these samples to the analysis?
So, finally, how many cells are discarded?

We didn't re-sequence more cells because it is expensive and we had enough for our analysis. We sequenced 92 haploid cells, and after discarding control cells and cells that didn't pass quality filters, we were left with 75 S-phase single cells. Similarly, for diploid cells, we started with 107 cells and 71 S-phase cells remained.

These numbers have been made more clear by the following changes in the text:

At the end of these data processing steps, 75 haploid S-phase cells had passed all quality controlTo

At the end of these data processing steps, 75 out of 92 haploid S-phase cells had passed all quality control.....

At the end of allele-specific parsing, 71 diploid S-phase cells had passed all quality controlTo

At the end of allele-specific parsing, 71 out of 107 diploid S-phase cells had passed all quality control.....

Paragraph #4

in fig1b, I would like to see a G2 profile and a replis-seq classical profile

The classical profile was in Figure 1b and the G2 profile would look exactly like the G1.

6. paragraph #5

Is it possible to have a box plot of distribution of the signals for each chromosome, because if the signal range change from a chromosome to other, you introduce a bias during binarization.

We have now added a boxplot (Supplementary Fig. 3) that shows the distribution of population replication timing values and distribution of copy number signal for single cells for each chromosome. However, while the signal range of all chromosomes are generally similar, we do not expect them to be the same. Some chromosomes have more late regions than others. Consequently the distribution of copy number signals will be lower for those chromosomes.

Why you remove outlier bins with a segmented value outside of +/- 2? and how many you removed?

These outlier bins weren't removed. These bins were only removed while calculating the threshold for binarization using Manhattan distances as described in the methods. We believe these are artifacts of single cell sequencing or sometimes amplifications or deletions that were too small to be identified by segmentation algorithms. The percentage of these outlier bins were very small (~0.1% of bins on average). 67 cells had at least one outlier bin. We have now added this quantification to the methods section.

You choose 100 equally spaced thresholds, why? Could you perform your Manhattan graph with 10, 50, 500 thresholds and indicate us which is the best?

The number of thresholds was an arbitrary choice that ensured more than adequate resolution. The "Manhattan graph" gave rise to a very predictable smooth curve. Adding more thresholds would have negligible effect on the value of the threshold and would increase processing time.

I calculated that you removed 20% of haploid cells, is not it too much?

Single cell genomics is an emerging field and technically challenging. Therefore we removed any cells that didn't pass several stringent quality controls. Our first series of experiments were using diploid cell and ~66% of cells we analyzed passed all quality filters. We improved technical aspects during subsequent trials and achieved ~82% success rate with the haploid cells.

7. paragraph #6

You mentioned region with low density. How many? Distribution for each chromosome? Which type of regions, more in Early or in late or same proportion ?

The point was to remove regions that have low reads after homologue parsing, not to study the distribution of SNPs between castaneus and musculus. The reader or reviewer could derive this for themselves if they are interested.

The minimum number of reads is 40 chosen with an empirical determination. Are you sure that in your case it is adapted?

The rationale is justified in reference #18, as cited (previously #14).

I calculated that you removed 35% of single diploid cells, is it not too much?

We kept only cells that had high quality data and they were sufficient to perform the described analyses. We did not want to lower the quality of the data.

In general, I would like to see the same analysis with 10 single cells, 500 and 1 000. It is possible that with more single cells we can see more variation from cell-to-cell and between homologous chromosomes

Analysis of 1000 cells would cost over \$40,000 USD.

8. paragraph #9

Again, can you determine the real S-phase length in your experiments?

This is similar to the comment above – S phase length is not important here, what is important is the percentage of DNA replication, which internally controls for the position of each cell in S phase.

9. paragraph#10

I would like to know if the homologue variations are more in Early or in late or in the same proportion.

This is one of the major conclusions of the paper – the variation is independent of early or late replication.

10. last paragraph

you written “within +/- 15%” How do you measure that ?

This is the T_{width} calculation shown in Figure 3a (previously figure 2d), so we have added “(Fig. 3a)” to the sentence.

In the sentence “overall, our results show...” there is two “THAT”

Thank you. It has been corrected.

In method

For data correction and smoothing, you calculated the mean of 5 G1 and 1 G2. I would have preferred 5 G1 and 5 G2.

We only had 1 G2 cell just as a proof of concept that both G1 and G2 have relatively uniform DNA content.

In Removing outlier cells paragraph, you indicate “some cells have low similarity...” How many exactly ?

19 cells for diploid; 5 cells for Haploid. This is now added to the method section where it is described.

Reviewer #2 (Remarks to the Author):

The authors present a detailed assessment of cell-to-cell variability in replication timing by utilizing high-quality single-cell DNA sequencing acquired from S-phase cells. Overall the manuscript is very well written and clear. The comparison to existing ensemble datasets suggest that the methods and analyses are robust, providing confidence in the quality of the study. The conclusions drawn from the analysis are well supported and justified.

In the second paragraph of the main text (pg4 top) where the authors briefly describe their workflow, they state that “Read counts were normalized by dividing the coverage data of each single-cell by the coverage of the G1 and G2 control cells.” This statement implies no normalization for variable read counts for each cell or binning over windows was used, which is not the case when reading the methods (which are described well). Changing this sentence to suggest what was actually carried out would add clarity. (eg “Windowed read counts per million total reads for each single cell were normalized by dividing...” etc...

We have now added the following sentences to the main text to make it clear:

“Read counts of all cells were converted to reads per million to control for variable sequencing depth.”

Also “single cell” is a noun, “single-cell” is an adjective. In most instances it is used correctly, but some may have been missed or auto-corrected which happens frequently in my experience

Thank you. We have now corrected these instances.

Figure 1c: A label for what the heatmap scale is showing should be added instead of having to root through the legend. Similarly, there should be scale bars on the bottom portion.

Done

I really appreciate the binarized analysis strategy. Very clear and seems to work well.

Thank you.

The authors attempt to investigate homologue-specific variability within the same individual cells, but appear to have been unable to due to the low haplotype-resolved read counts. (at least not in a meaningful way). While I appreciate this analysis, the power was not really there, so it may be worth removing the details from the main text and simply stating that it was attempted but coverage was too low and refer to the methods section. After continuing reading, the authors then come back to homologue-specific differences based on the domain-specific variability and see little difference. So they were able to say something meaningful with that analysis. Perhaps consolidating the homologue-specific analysis to this one section would help with the flow of the manuscript instead of earlier on where it appears nothing came of it then looping back to it.

We are sorry that we gave the impression early in the manuscript that the haplotype-resolved read counts were too low for analysis. We believe that the offending sentence in the prior version of the manuscript was: “However, despite the high density of SNPs, homologue parsing nonetheless results in significant loss of coverage, particularly in regions of low SNP density.” What we meant by a significant loss of coverage was that we lost low SNP density regions, the number of which depended upon how many total reads we had per cell and we wanted to describe how we filtered for high confidence regions. This analysis was immediately put to use. To clarify this, we have made the following changes:

The above sentence was changed to: “Since regions of low SNP density have fewer sequencing reads after parsing, and we had variable total reads per cell, we filtered for those segments with high confidence of replication asynchrony as follows.”

We also changed the first sentence of the next paragraph to (phrase added underlined) “The combined haploid and homologue-parsed diploid binarized data..” to re-emphasize that the parsed data was used.

Figure 2d-g could be lined up a bit better or even made a bit larger – these are plots that do a lot to convey important information.

To make the figures 2d-g more visible, we have now split figure 2 into two separate figures.

Fig 3b include a title for the color scale bar.

Done

P13 typo: “pairs of bind” should be “pairs of bins”

Thanks for catching, now corrected.

Correlation with HiC compartments is great.

Thank you.

Some of the conclusions are written in a contradictory way – ie “highly conserved, yet stochastic” but “far from random”. I understand the conclusion that is being made, but it should be broken down into what you mean by the stochastic component (ie domains do not always activate at the same time/order in a replication origin selection context) and that when you refer to random you are implying no specified program, but genome-wide random which is not the case because actual timing is quite close.

We have now avoided all of these adjectives and simply stated: “The results support a model in which intrinsic variability and extrinsic cell-to-cell variability are similar, regardless of the timing or chromatin state of each domain, with most individual domains replicating within +/-15% the length of S-phase from their scheduled time.”

Reviewer #3 (Remarks to the Author):

General

This submission provides a novel perspective of chromosome structure, probing into the flexibility of the replication timing program in the context of large, functionally distinct chromatin compartments. The paper reports the development and implementation of single-cell based analyzes of replication timing domains, demonstrating that the boundaries of chromatin regions that share the time of replication at a single cell level were similar to the boundaries of active and inactive chromatin compartments. Importantly, mapping the extent of genome

duplication in single cells, and analyses of the extent of variations in single cell replication timing, allow an accurate interpretation of population-based replication timing data.

The results reported in the paper were obtained using novel methodologies to measure DNA copy number in individual cells. The studies were performed with sufficient resolution to distinguish between chromosomal zones of replicated and unreplicated DNA, providing a snapshot of the stage of replication completion (corresponding to cell cycle stage) in single cells. The paper provides compelling evidence that the replication timing program could be inferred from single-cell based measurements. This is an important methodological development, likely to provide tools that will considerably advance the field of DNA replication.

The paper is written very clearly, presenting state-of-the-art experiments with appropriate controls. Single-cell copy number variation data are described with sufficient detail. The manuscript will be improved by the inclusion of a better description of the population-based (Repli-seq) data. This is important, because the relationships between the single-cell data and population-based data are critical for the variability analyses. Some other minor points require clarifications. Overall, however, this report represents an important contribution to the field.

Comments and minor suggestions:

1. The manuscript in its current form describes the methodology and primary observations for single-cell analyses in detail (for example, supplementary Figure 4), but the parallel report for Repli-seq does not provide the same level of detail. The primary reference for the replication timing data is an unpublished BioRxiv protocol. Although population-based replication timing analyses are less novel than the single-cell data, the comparison is important and both sets of data should be presented with sufficient detail. In addition, the current analysis relies on a novel variation of the Repli-seq technique, and the paper would benefit from a more detailed description of the results.

A detailed description of the repli-seq protocol was cited as a Biorxiv paper and has recently been accepted for publication by Nature Protocols and we will reference this paper as soon as the online version is released. However, we appreciate the reviewers' comment and have provided a brief description of the salient steps in the methods section.

2. Related to the above, Page 7, last paragraph: "These domains correspond to early replicating CTRs identified in population-based replication-timing profiles". A citation, a reference to a figure, or a URL with the population-based data should be provided here.

A citation has been added.

3. To allow a full evaluation of the results, the paper should provide URLs for data access (single-cell and population based) in public depositories.

The GEO accession number and password is provided under data access.

4. Analyzing the extent of variation based on the ratios of replication timing in the two mouse species is a clever and informative approach. However, it would be good to include an analysis of the general parameters underlying the variations reported in the paper (for example, it would be beneficial to report the size distribution of replication domains (CTRs and TTRs) in the two homologue-parsed chromosomal sets).

We have done a thorough analysis of the species differences in replication timing in these same hybrid cells, but this is the topic of a separate manuscript. Overall the two genomes differ significantly across only 12% of the genome. We have now added a statement to that effect, referenced as “unpublished observations”. We have also provided the reviewer with a copy of this manuscript.

“Population Repli-seq has revealed high conservation of replication timing profiles between the homologues with only 12 % of the genome showing detectable genome-specific variation (unpublished observation)”

5. Related to the above, how does the variation in population-based replication timing domains in the current study relate to variations in chromatin compartments calculated from Hi-C in the hybrid and the parental species? Also, how does the extent of variation between homologues in the current study compare to the variation obtained in previous reports of replication timing variation (for example, in humans using a phased chromosome approach: Mukhopadhyay et al pubmed id 24787348)?

If we understand the reviewers’ question correctly, an analysis of stochasticity in Hi-C compartments would require an extensive analysis of Peter Fraser’s single cell Hi-C data, which would seem beyond the scope of this manuscript describing single cell replication timing.

As for the Mukopadhyay et al paper, this topic is also covered in the manuscript mentioned in point 4, which we have provided.

6. The manuscript in its current form reports data from hybrid cells on page 4, yet the benefits of using those hybrids to the study of differential replication timing variations are described in detail in page 6. It would be better to move the text

from the last paragraph on page 6 to an earlier point in the manuscript before reporting the results.

Actually, we tried to put the benefits of the hybrids at the top of page 4, but apparently it was not clear enough. We have modified the sentence as follows:

“Moreover, to separately evaluate the extent of extrinsic (cell-to-cell) vs. intrinsic (homologue-to-homologue) variability in replication timing, we examined both the differences in replication timing between haploid H129-2 mouse embryonic stem cells (mESCs) and the differences between maternal and paternal alleles in diploid hybrid *musculus* 129 × *Castaneus* mESCs that harbor a high single nucleotide polymorphism (SNP) density between homologues, permitting allele specific analysis.”

7. Legend to Figure 1: “population-based BrdU-IP”: if this is the same technique referred to as “Repli-Seq” in the text, it would be good to use a consistent term throughout the manuscript.

We have now changed this to “Repli-seq”.

REVIEWERS' COMMENTS:

Reviewer #1 (Remarks to the Author):

I have read the remarks and corrections made by the authors for their revised manuscript. Their answers have fully convinced me and I will support the choice of the editors if they decided to accept this article in Nature communication

Reviewer #2 (Remarks to the Author):

My primary concern with the manuscript was with some of the language and terms that sometimes seemed conflicting without a detailed read through or additional background knowledge. The authors have removed the conflicting statements and I believe it has resulted in a much clearer manuscript. All other comments I had were also addressed. I believe the manuscript is suited for publication in its revised form.

Reviewer #3 (Remarks to the Author):

The authors have addressed all my concerns and suggestions. My only additional suggestion, if compatible with the journal's policies, is to consider a citation of the associated preprint describing allele-specific control of replication timing. This preprint was discussed in the reviewers' correspondence and is newly available in BiorXiv.